# Deep neural network inference on an integrated, reconfigurable photonic tensor processor

Lennart Meyer [1], Jelle Dijkstra [1], Simon Tebeck [1], Liam McRae[1], Niklas Bahr[1,2], Daniel Steinmeyer [2], Sergey Koptyaev[3], Johana Bernasconi [3], Nikolay G. Pavlov [3], Maxim Karpov [3], John D. Jost[3], Wolfram Pernice [1] & Frank Brückerhoff-Plückelmann [1] ✉

Artificial neural networks set the pace in machine vision, natural language processing, and scientific discovery, but their performance depends on fast and efficient tensor computations. Analog photonic systems are a promising alternative to digital electronics because they enable ultra-fast, low-latency computing while avoiding capacitive charging losses and electrical crosstalk. Here we present a photonic tensor processor for deep neural network inference, integrated into a standard 19-inch rack unit with a high-speed electronic interface to PyTorch for seamless hardware deployment. The processor implements an all-optical crossbar with nine inputs and three outputs for parallel intensity-based accumulation of weighted signals. Fabricated in imec's iSiPP50G silicon photonics platform, the chip integrates electro-absorption modulators and photodiodes for scalability and compatibility with high-volume manufacturing. An integrated self-injection-locked microcomb provides stable multi-wavelength carriers. We demonstrate inference on MNIST and CIFAR-10 with 98.1% and 72.0% accuracy, highlighting a compact, reprogrammable platform toward scalable high-speed optical AI accelerators.

Artificial intelligence is becoming increasingly central to science, industry, and society. As models grow in size and complexity, they place ever greater demands on computational infrastructure[1,2]. A major driver of this demand is tensor operations, a fundamental operation underpinning nearly all layers of modern neural networks. At the scale of modern AI models, inference requires hundreds of billions to trillions of multiply-accumulate (MAC) operations, making tensor processing the dominant contributor to latency and energy consumption[3–8].

Analog computing is gaining renewed interest for accelerating AI by performing linear algebra operations directly in the physical domain as signals propagate[9,10]. In the electronic domain, mature crossbar arrays with memory cells which store weights at each junction exemplify this approach[11–13], with IBM recently showcasing a 64-core in-memory chip capable of deep network inference[14]. Alternatively, Lightmatter's and Lightelligence's processors employ hybrid photonic–electronic architectures, leveraging light's intrinsic parallelism and speed to push computational performance beyond conventional limits[15,16]. By combining optical input modulation with electronic accumulation, these hybrid systems achieve near-digital precision across demanding AI workloads and ultra-low latency in optimization tasks. Photonic approaches bring exciting advantages well-known from telecommunication, such as low latency and low propagation loss. Thus, the possibility to fully leverage these properties through purely optical systems has inspired many different approaches to all-optical computing, including phase-change-material-based in-memory computing[17,18], time-wavelength interleaving[19], multiplexing across multiple degrees of freedom[20–23],

[1]Kirchhoff-Institute for Physics, University of Heidelberg, Heidelberg, Germany. [2]Volkswagen AG, Wolfsburg, Germany. [3]Enlightra, Renens, Switzerland. ✉e-mail: frank.brueckerhoff-plueckelmann@kip.uni-heidelberg.de

diffraction[24,25], coherent MZI meshes[26,27], and utilizing partial or incoherent light sources[28–30]. While several demonstrations achieve striking (theoretical) computational powers, many are hard-wired for specific operations, require abstract problem mappings, or face integration and system-level constraints.

Here, we present an integrated photonic tensor processor (PTP) for deep neural network inference, where the linear tensor operations are performed all-optically. The PTP adds optical intensities from incoherent inputs and thus requires only an output photodiode array sized to the output dimension. Electro-absorption modulators provide high-speed input and weight modulation and a one-to-one mapping from model weights to drive levels, enabling arbitrary tensor operations without auxiliary transformations except scaling. Packaged as a rack-mounted system with electronic I/O, calibration procedures, and PyTorch integration, the PTP executes pretrained networks on photonic hardware without chip-specific retraining.

## Results

### System architecture and operation

Our PTP is based on a silicon on insulator (SOI) photonic integrated circuit (PIC). The PIC is fabricated on imec's iSiPP50G silicon photonics platform and implements an incoherent optical crossbar array with integrated electro-absorption modulators (EAMs) and photodetectors. The SOI chip contains EAMs to encode input vectors and to set matrix weights through changes in transmission, while on-chip photodiodes convert the results of the optical matrix-vector multiplication into electrical current. A schematic of the PTP architecture is shown in Supplementary Fig. 3.

In order to embed the photonic system within a computer-addressable electronic framework, we employ a ZCU216 RFSoC with a high-speed field programmable gate array (FPGA) to drive and read out the PIC. The integrated on-board RF-DACs run at 4 GS/s for EAM modulation and the RF-ADCs at 2 GS/s for readout. An external host connects over Ethernet to a Jupyter server on the RFSoC's CPU for interactive control, while the CPU generates the vectors and weight sequences. The CPU then hands these values to the FPGA, which conditions and routes them. High-speed streams are directed to the RF-DACs (EAM inputs) and slower settings to multi-channel DACs that program the crossbar. Reprogramming the full weight array takes 62 ms, whereas streaming inputs and digitizing outputs operate continuously at the RF-DAC/RF-ADC sample rates. Transimpedance amplifiers (TIAs) convert the PIC outputs to voltages, and the RF-ADCs digitize the voltages. The FPGA can decimate the samples before the CPU returns results and metadata to the host. TIAs and weight DACs sit off-board and connect directly to the RFSoC.

We wire-bond the photonic chip to a custom carrier printed circuit board (PCB, Fig. 1a), which we insert into a system board using high-speed RF cables. This allows straightforward physical integration and provides electrical connectivity. The entire system is housed in a standard 19-inch rack. Slow control signals, such as bias voltages ranging from −10 to 10 V, are available via dedicated DAC channels. Figure 1b shows a schematic overview of the system.

In order to optically drive the system, we employ a fully packaged and low-noise self-injection-locked (SIL) microcomb based on a high-Q $Si_3N_4$ microresonator[31–35] as the input light source. We utilize individual comb lines as input carriers from a single comb source with a fixed spectral spacing, avoiding the need to manually tune and stabilize multiple laser sources. The microcomb provides a 485 GHz free-spectral range (FSR) and delivers about -11 mW total output, of which we tap 5% for monitoring (Fig. 2c). We demultiplex the comb lines and route the individual carriers to the input grating couplers. EAMs encode vector entries at the inputs onto the different wavelengths using a four-samples-per-symbol scheme with differential DAC drive, and EAMs at each cell store the matrix weights. We compensate for the nonlinear response and wavelength-dependent extinction ratios

(Fig. 2d) during weight calibration, which stabilizes effective weight accuracy across carriers. PIC-integrated SiGe photodiodes convert the accumulated optical outputs. We operate the photodiodes at 3 V reverse bias, chosen after observing diminishing responsivity gains beyond 3 V (Fig. 2e). Optical packaging influences the practical power budget. Careful four-axis alignment yields about 2.5 dB insertion loss, and adhesive bonding introduces roughly an additional 1.5 dB along with a wavelength shift (Fig. 2f). We account for these effects in per-channel/per-carrier power allocation and during calibration. Together, carrier power, calibrated EAM nonlinearity, photodiode biasing, and packaging-aware power allocation enable stable, parallel optical multiply-accumulate execution on chip and define the operating envelope for further system scaling.

### Weight programming

We employ a dedicated calibration routine to set the system accuracy by mapping target weights to modulator drive voltages using measured transfer functions. We further apply crosstalk compensation and a global output rescale to correct inter-channel coupling and restore a consistent gain. The Supplementary Note 9 details the complete procedures. We realize signed weights with a balanced readout scheme in which each logical weight is the transmission difference between a main row and a reference row. This symmetric encoding centers the operating point and implements signed multiply–accumulate behavior. The primary performance metric is the statistical output error for matrix vector multiplications, $\epsilon_{MVM}$, defined as the normalized difference between the optically processed results $\bar{y}_k$ and ideal digital references $y_k$.

$$\epsilon_{MVM} = \frac{\langle ||y_k - \bar{y}_k||_2 \rangle}{\langle ||y_k||_2 \rangle} \tag{1}$$

In order to improve overall accuracy, we optionally implement averaged measurements at the cost of decreasing throughput and increasing latency. The MVM error initially decreases with the root of the number of averages, corresponding to a dominantly stochastic error, e.g., from the thermal noise of the transimpedance amplifier. The single-shot, low-latency mode exhibits an error of $(19.4 \pm 0.5)\%$, while averaging four times reduces the error to $(10.9 \pm 0.3)\%$ in a higher-precision mode. Additional averaging yields diminishing returns and approaches a systematic noise floor near 3% as shown in Fig. 3a. This noise floor is mostly attributed to a non-perfect weight programming, also see Supplementary Note 4. Figure 3b compares measured and ideal outputs. The data points cluster tightly around the diagonal, and the error histogram follows the anticipated one over the square root of the averaging factor scaling, indicating that noise is predominantly stochastic. Figure 3c reports absolute weight error versus target weight. We infer effective analog weights with a least-squares fit from measured inputs and outputs. The mean absolute error is below 5%, with larger deviations at higher magnitudes that reflect the asymmetric nonlinear response of the modulators. Finally, we evaluate the system stability over a measurement period of 120 min. For both the low-latency and precision modes, the MVM error stays within 1.1% of the respective mean value, enabling reliable analog computation over long timescales.

### Deep neural network inference

To benchmark the system, we develop and evaluate two convolutional neural networks on the photonic tensor processor platform. The first model is a compact baseline network with two convolutional layers followed by a fully connected output layer for 10-class classification. We use this model exclusively for MNIST. The second model is a slightly deeper architecture optimized for CIFAR-10, featuring four convolutional layers before the classification layer. Tables 1 and 2 detail both networks. During inference, the photonic hardware

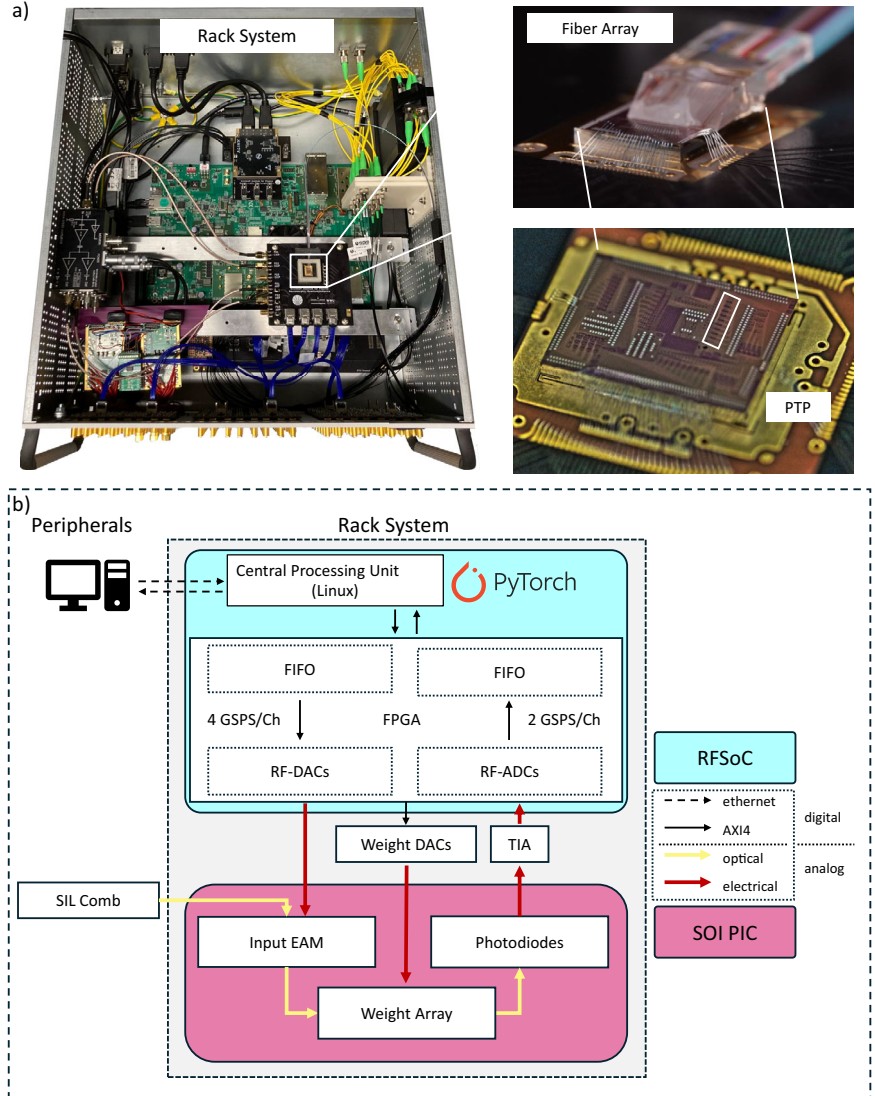

**Fig. 1 | Photonic tensor processor architecture. a** Photonic Hardware accelerator in a 19-inch rack housing, RFSoC evaluation board, chip on carrier PCB, TIAs, weight DACs, and optical splitters. Close-up of the wire-bonded PIC with glued fiber array, and a top-view image of the PIC. **b** Schematic overview of the setup and data flow.

performs the convolution and fully connected layers, and the remaining operations run digitally, translating to 97.5% and 99% optical workload, respectively.

We pretrain both models digitally using PyTorch, employing stochastic gradient descent with momentum for initial training, followed by Adam for hardware-aware fine-tuning. To improve model robustness, we apply cross-entropy loss and data augmentation. After training, we perform fine-tuning over 50 epochs using AIHWKit-Lightning[36], incorporating hardware-aware weight noise (fixed at 5%) and output noise (10 and 20%) to reflect the system's measured behavior. After fine-tuning, we deploy both models on the photonic system for inference. As shown in Fig. 4, the smaller network achieved a classification accuracy of approximately 98% on the full MNIST test dataset in precision mode, and 91% in low-latency mode. For CIFAR-10, we use the more expressive model in precision mode, since CIFAR-10 classification is a much more complex task with less noise tolerance. Photonic inference on a subset of 400 images reaches an accuracy of 72%.

The larger drop in accuracy during photonic inference for CIFAR-10 is mainly driven by the larger layer sizes and, consequently, the higher dimensionality of the underlying matrix-vector multiplications. Because the PTP has a finite matrix size, we implement these MVMs via

tiling, i.e., we decompose an MVM into $K$ partial MVMs and accumulate their outputs digitally. If the partial outputs are uncorrelated and the MVM error is uncorrelated and stochastic, both the signal and the absolute error scale $\sim \sqrt{K}$, such that the relative error remains approximately constant. In contrast, correlated, systematic errors scale as $\sim K$, causing the relative error to increase with the amount of tiling. The PTP system is dominated by stochastic noise, as shown in Fig. 3a, and thus supports tiling. However, a small systematic error component is present and can become dominant for a large amount of tiling. In the CIFAR-10 network, the fully connected classifier has the largest input dimension. Executing only this layer digitally increases the accuracy by 6%, also see Supplementary Note 3.

## Discussion

The fully programmable analog PTP executes tensor operations with trained weights and stable photonic I/O. A calibrated mapping from target weights to modulator voltages, including inter-channel crosstalk compensation, enables accurate and repeatable tensor operations. End-to-end inference on MNIST and CIFAR-10 validates practical utility under real-world conditions. With hardware-aware fine-tuning and moderate averaging, MNIST remains near a digital baseline, while CIFAR-10 reflects tighter noise tolerance yet

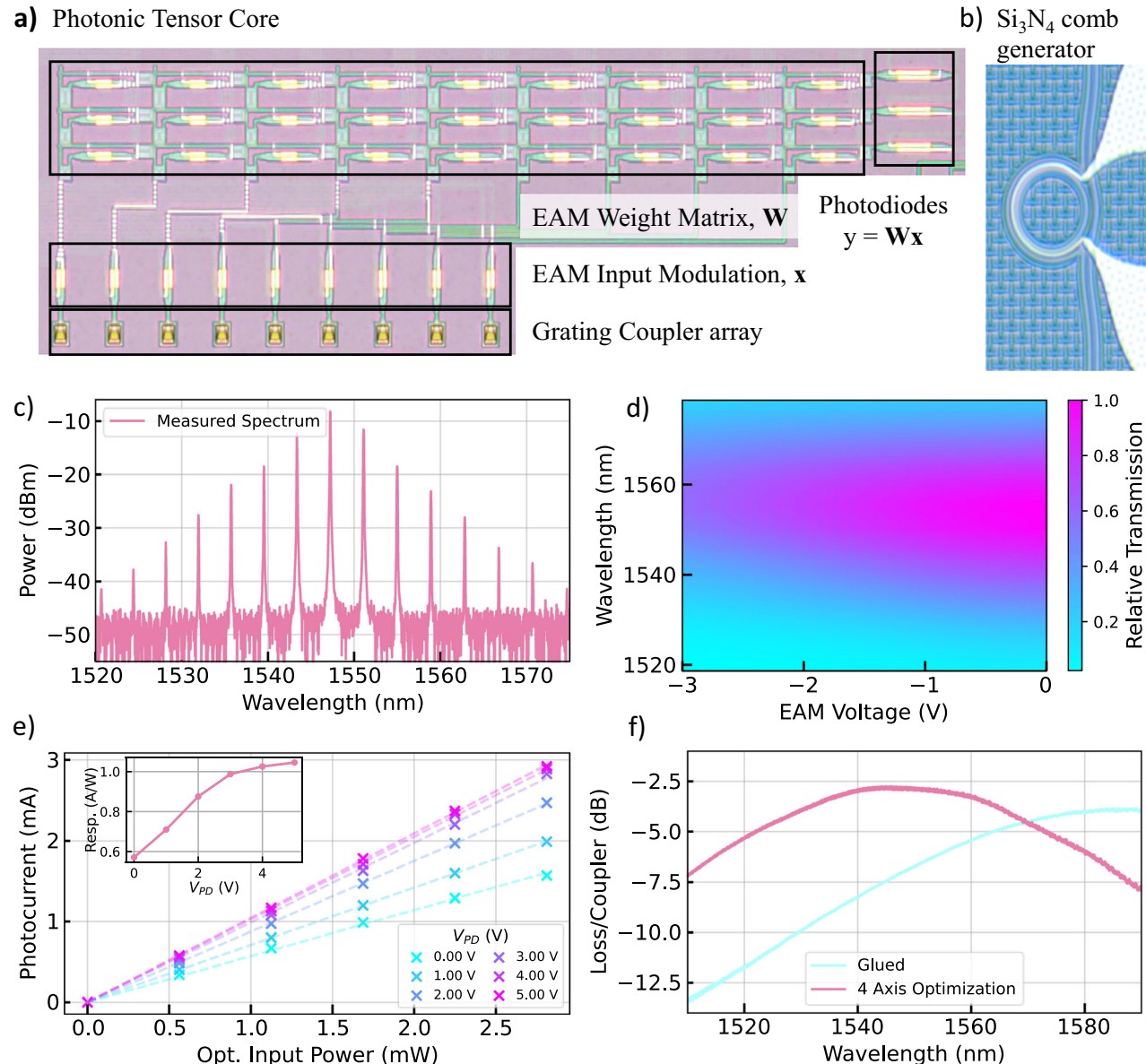

**Fig. 2 | Characterization of individual components. a, b** Chip-level pictures of the Photonic Tensor Processor and high-Q $Si_3N_4$ microresonator used for microcomb generation. **c** Optical spectrum of the self-injection-locked microcomb with a FSR of 485 GHz. **d** Wavelength- and bias-voltage-dependent transmission of the EAMs. We operate the modulators around a bias voltage of 2 V in a linear regime. **e** Photocurrent vs optical input power for different bias voltages with the corresponding responsivity curve. We operate the photodiodes with a bias voltage of 3 V. **f** We measure transmission of a test structure before and after gluing of the fiber array. Due to the glueing and shrinkage, the peak transmission drops from −2.7 to −3.8 dB.

maintains useful task performance. Beyond component- or core-level demonstrations[20,22,28,37], this work shows a deployable inference system, integrating calibration, electronic I/O, and a PyTorch interface that runs pretrained networks without chip-specific retraining. By explicitly measuring the hardware noise statistics but using a Gaussian error abstraction, training is decoupled from a particular chip instance, supporting portability across devices and enabling scalable deployment. This establishes a functioning optical hardware accelerator that realizes the same operation class as electronic accelerators with programmable linear layers.

Because the system is analog, it is subject to noise, distortions, and device variability. Deep networks are robust to reduced precision, often around four bits[38–40], and even lower with mixed precision[40–43], which motivates our tolerance targets. We measure an accuracy-latency trade-off via temporal averaging. In a single-shot and lowest latency pass, the mean MVM error is about 20%.

Averaging by four reduces this to about 11%. The error is mostly stochastic and decreases approximately as the inverse square root of the number of averages. Therefore, improving the signal-to-noise ratio, for example, by reducing loss in individual components, will further boost system performance.

A key benefit of photonics is that it sidesteps capacitive-charging limits that dominate large electronic crossbars and interconnects. Because signals propagate optically within the tensor processor, we avoid tensor-size-dependent charge/discharge penalties that bound electronic latency[44–46]. In our incoherent intensity-accumulating architecture, a single optical wavefront executes the full tensor operation, so optical transit and I/O bandwidth limits set the latency rather than the tensor dimensions themselves. One modulation interval can realize a single-cycle tensor operation, assuming loss and bandwidth remain in budget. Static-weight photonic systems, such as diffractive or fixed-weight designs, can deliver remarkable throughput

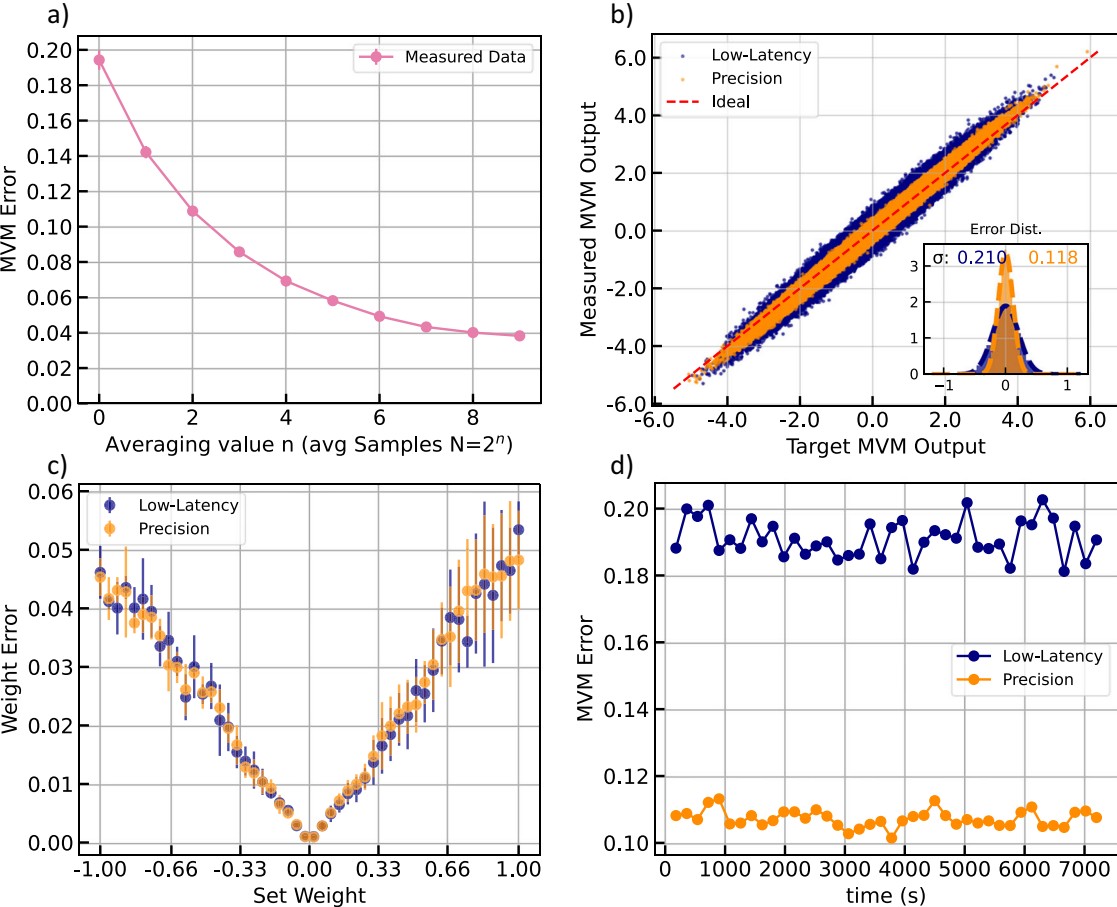

**Fig. 3 | System evaluation of the PTP. a** MVM error vs. averaging. The decreasing error suggests mainly stochastic error sources down to a noise floor of ~3%. **b** Measured MVM results vs. target MVMs with error distribution (inset). The measured data points lie close to the ideal diagonal. Lower errors for the precision setting of the PTP vs. the fast mode are visible. **c** Weight error vs. set weight. We estimate the actual weight of the analog system using a least-squares regression from the measured MVM output and input matrix. **d** MVM error of the two PTP settings over time.

and efficiency, yet they typically implement a fixed transform with a limited set of operations[21,22,24,47]. Here, we program arbitrary trained weight matrices electrically, bringing photonic MVM close to the flexibility of electronics while preserving the latency advantage. To achieve this flexibility, we separate the time scales of weights and activations. We operate the processor in a weight-stationary inference regime, keeping the weights constant over long sequences of input vectors to minimize digital data transfer, while inputs and outputs are streamed continuously. The reported 62 ms corresponds to a full array update of all weight control voltages and is dominated by control path overhead rather than by intrinsic device limits. In contrast, input activations are modulated at 4 GSPS and outputs are sampled at 2 GSPS, so sustained throughput is set by system-level input–output bandwidth rather than weight updates. We choose these sampling rates as a trade-off because higher ADC sampling rates reduce conversion efficiency[48–50].

In the present prototype, energy efficiency is dominated by system-level electronics, in particular DACs, ADCs, and TIAs, rather than by the optical core itself. Using a projected total power consumption of 2.5 W for the $9 \times 3$ configuration under continuous streaming at an effective 1 GHz symbol rate, the system achieves 27 GMAC/s, corresponding to 0.022 TOPS/W. For comparison, NVIDIA reports an INT8 throughput of 3958 TOPS (with sparsity) at 700 W (H200 SXM)[51] corresponding to ~5.65 TOPS/W. Analog electronic accelerators achieve 1.74–9.34 TOPS/W depending on task and operating point[52], and analog electro-optic hybrid accelerators report about 0.82 TOPS/W[15]. Importantly, the efficiency is expected

to increase with throughput. For an $n \times n$ tensor core, the number of MAC operations per symbol scales as $n^2$, whereas the dominant converter and amplifier overhead scales with the number of analog channels, $\sim n$.

Scaling core size, parallelism, and bandwidth increases the compute performance of the PTP. Crossbar arrays up to $32 \times 32$, without an electro-optic interface, have been demonstrated[22], and multiplexing across additional degrees of freedom, such as wavelength division multiplexing, enables processing several MVMs with a single core in parallel[53]. Increasing electronic interface bandwidth enables proportionally higher streaming rates, leveraging the large bandwidth of photodiodes and electro-absorption modulators[54]. However, efficiency eventually decreases because ADC conversion efficiency degrades at higher sampling rates. As one perspective, a $32 \times 32$ core with 4 wavelength channels at 1 GHz computes 8.2 TOPS. In addition to scaling the PTP, hardware-software co-design provides a complementary lever on the model side, for example, via grouped and depthwise-separable convolutions that reduce effective matrix sizes and map naturally onto smaller photonic tiles.

Latency is a principal strength of our PTP, which performs linear matrix vector multiplications in the optical domain by the propagation of optical fields within the integrated circuit. Unlike systolic arrays, the full matrix vector multiplication is executed within a single symbol period, without deep pipelining, reducing the time to first result. For the sub-millimeter scale photonic circuit used here, the optical propagation time is negligible compared to the symbol duration. Deploying additional degrees of freedom for

**Table 1 | Network architecture for MNIST classification**

| Type/Stride | Filter shape | Input size |
|---|---|---|
| Conv/s1 | $3 \times 3 \times 1 \rightarrow 16$ | $28 \times 28 \times 1$ |
| ReLU | - | $28 \times 28 \times 16$ |
| MaxPool/s2 | $2 \times 2$ | $28 \times 28 \times 16$ |
| Conv/s1 | $3 \times 3 \times 16 \rightarrow 32$ | $14 \times 14 \times 16$ |
| ReLU | - | $14 \times 14 \times 32$ |
| MaxPool/s2 | $2 \times 2$ | $14 \times 14 \times 32$ |
| FC | $7 \times 7 \times 32$ (flattened) $\rightarrow 10$ | 1568 |

**Table 2 | Network architecture for CIFAR-10 classification**

| Type/Stride | Filter shape | Input size |
|---|---|---|
| Conv/s1 | $3 \times 3 \times 3 \rightarrow 32$ | $32 \times 32 \times 3$ |
| BN + ReLU | - | $32 \times 32 \times 32$ |
| Conv/s1 | $3 \times 3 \times 32 \rightarrow 64$ | $32 \times 32 \times 32$ |
| BN + ReLU | - | $32 \times 32 \times 64$ |
| MaxPool/s2 | $2 \times 2$ | $32 \times 32 \times 64$ |
| Conv/s1 | $3 \times 3 \times 64 \rightarrow 64$ | $16 \times 16 \times 64$ |
| BN + ReLU | - | $16 \times 16 \times 64$ |
| Conv/s1 | $3 \times 3 \times 64 \rightarrow 128$ | $16 \times 16 \times 64$ |
| BN + ReLU | - | $16 \times 16 \times 128$ |
| MaxPool/s2 | $2 \times 2$ | $16 \times 16 \times 128$ |
| Dropout | $p = 0.2$ | $8 \times 8 \times 128$ |
| FC | $8 \times 8 \times 128$ (flattened) $\rightarrow 10$ | 8192 |

data encoding can increase parallelism and throughput without increasing the single-cycle latency[20,22,23,28,55]. This low-latency regime is most attractive when one large or recurring layer sets the step time, including recurrent architectures such as LSTMs and Hopfield nets.

## Methods
### System setup
We deploy an Enlightra SIL-microcomb as a light source and split the optical intensity using an Agiltron splitter. An Ando AQ6330 optical spectrum analyzer monitors the low power output. We amplify the high power output using a PriTel LNHPFA-33. An FS FMU-D402160M Multiplexer filters the relevant wavelength channel, C23/C28/C33/C37/C42 of the ITU grid. We amplify C23 and C42 with a PriTel LNHPFA-33-Pre-Amp each and route all optical carriers through Thorlabs FPC562 Polarization controller to the PIC. We glue an SQS fiber array to the PIC with the photoresin Nanoscribe IP-S. Femto HSA-Y-1-60 TIAs interface the chip and convert the photocurrent. Two Analog Devices DC2025A-A boards provide voltage for the weight array of PTP. An RFSoC ZCU216 Evaluation Kit orchestrates the electronic interface. An Arroyo 6305 combo source and two Keithley SourceMeters (2450 and 2400) control the TEC, the laser diode, and two phase heaters of the comb, respectively.

### Device fabrication
We fabricate the PTP on imec's iSiPP50G silicon photonics platform during a multi-project wafer run (https://www.imeciclink.com/en/asic-fabrication/silicon-photonics-foundry-services). The design kit provides the monolithically integrated electro-optic devices we used. Directional Couplers for arbitrary splitting ratios were designed based on the coupling parameters of the design kit. The diced PIC is mounted on a custom printed circuit board. The PCB is fabricated using Eurocircuits defined impedance pool with 4 layers (https://www.eurocircuits.com/services/defined-impedance-pool/).

### Photonic matrix
We encode each input vector element $(x_1, \ldots, x_M)$ onto a different wavelength channel using the on-chip modulators. The matrix is implemented as a waveguide crossbar array[22] equipped with directional couplers that evenly distribute the optical power to all EAM cells. For an $M \times N$ matrix, the horizontal couplers have splitting ratios of $1/(N - j + 1)$ for column index $j$, and the vertical couplers have splitting ratios of $1/i$ for row index $i$. The matrix elements themselves are encoded in the EAM transmission. A frequency comb with a line spacing larger than the electrical bandwidth ensures interference-free detection of the summed optical intensities along the individual columns. The output power at each column photodiode corresponds to the inner product between the input vector and the respective kernel.

### Input encoding
We are using a zero-mean, four-sample, return-to-zero, alternating encoding, such that the TIAs see only changes around the optical bias. Our desired input symbol is

$$k_n \in [-1, 1]$$

We encode it into 4 DAC samples without a DC part, setting the operation/symbol rate of the system to a quarter of the DACs speed:

$$\boldsymbol{s_n} = [s_{4n+1}, s_{4n+2}, s_{4n+3}, s_{4n+4}] = [k_n, 0, -k_n, 0] \quad (2)$$

We drive the ADC at half the speed of the DAC and sample at the non-zero time slots:

$$\widetilde{k}_n = C(s_{4n+1} - s_{4n+3}) \quad (3)$$

With C being a constant of all involved device parameters (details in the Supplementary Note 7). The two nonzero samples produce opposite-signed fluctuations around the bias, so the AC-coupled output carries a signed value even though the optical power itself is non-negative. Normalizing with $\frac{1}{2C}$ recovers the encoded symbol

$$\frac{1}{2C} * \widetilde{k}_n = \frac{1}{2C} * C(s_{4n+1} - s_{4n+3}) = \frac{1}{2C} * C(k_n - (-k_n)) = k_n \quad (4)$$

### Weight error
The weight Error is defined as:

$$\epsilon_{weight} = \frac{||w - \widetilde{w}||_2}{\Delta w} \quad (5)$$

with $\Delta w = \max(w) - \min(w)$

With w being the desired weight and $\widetilde{w}$ the weight set by the PTP. For each target weight value and averaging setting, 1000 random input vectors were processed using the photonic hardware. The photonic output was used to reconstruct the actual implemented weights $\widetilde{w}$ via regression. From this, the weight error represents the deviation between the reconstructed and target weights. Repeating this procedure for all weight values and averaging settings yields the datasets of weight errors and reconstructed weights, which is shown in Fig. 3c.

### MVM Error
For the MVM vs averaging measurements in Fig. 3a, 20 independent runs were performed to assess how averaging affects the accuracy of photonic matrix–vector multiplications. In each run, 1000 random input vectors of dimension 10 were processed through a $10 \times 10$ photonic weight matrix. The averaging parameter was varied over 10 discrete levels. For every run and averaging setting, the photonic MVM

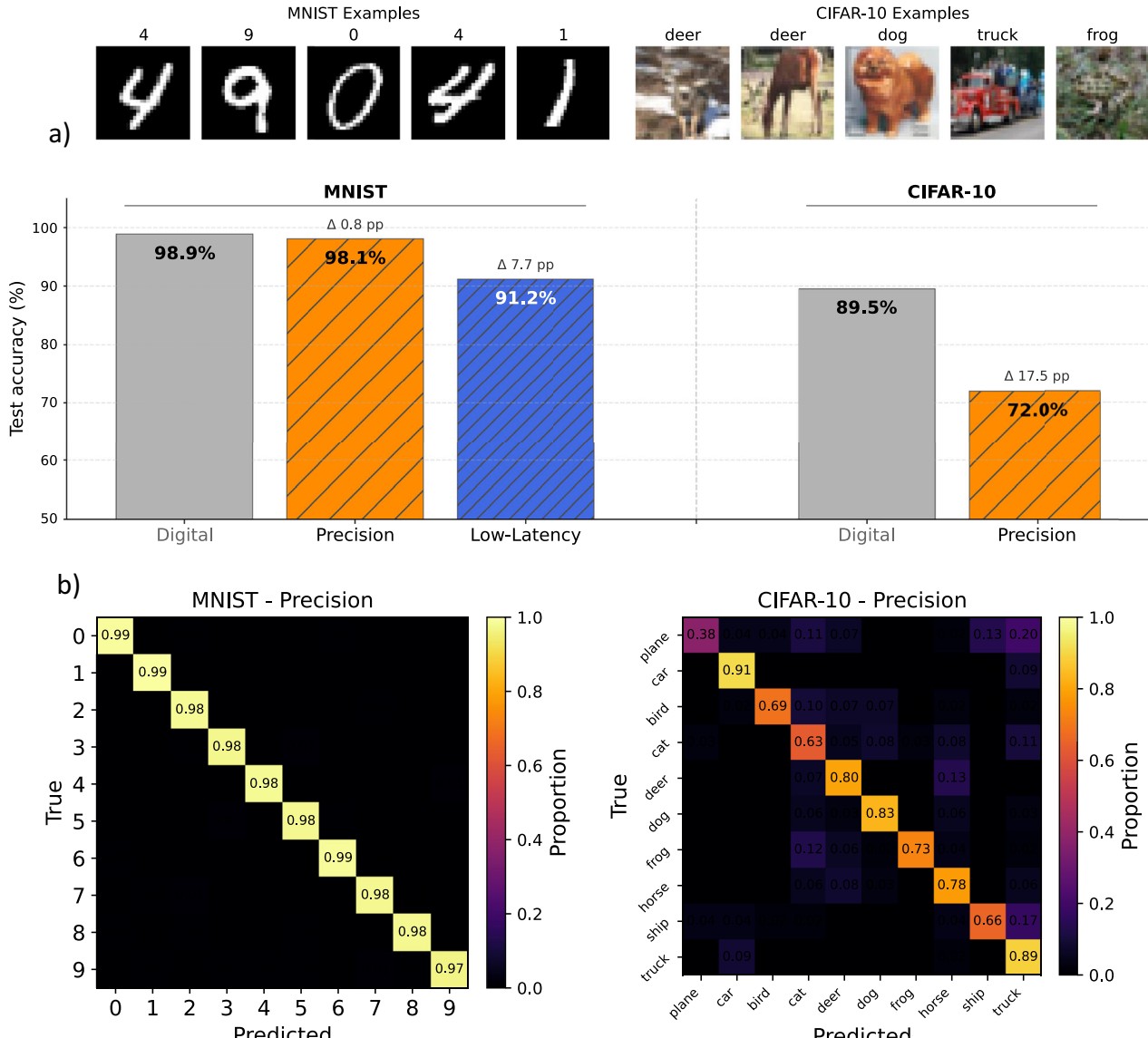

**Fig. 4 | Optical inference classification results. a** Measured test accuracy of both network architectures for different settings. The "mnist-net" on the left and the "cifar10-net" on the right. The accuracy of the digital networks in gray. Low-latency, precision mode of the PTP in orange and blue. **b** Confusion matrices for both datasets in the precision mode.

outputs were compared with digitally computed reference results to obtain the total MVM error from Eq. 1. All measured and reference output matrices (1000×10 per averaging level) were recorded to enable detailed statistical analysis of accuracy scaling with averaging.

To examine the statistical distribution of individual MVM output errors of Fig. 3b, the complete measured and reference datasets from 100 runs were evaluated at two averaging settings (low-latency and precision). Each run processed 1000 random input vectors of dimension 10 through a 10 × 10 photonic weight matrix, producing 1000 output vectors with 10 elements each. Every individual value within these output vectors—that is, each input–output product corresponding to one row–column multiplication result of the MVM—was compared to its digitally computed reference. The measured values were plotted against their targets to visualize overall fidelity, and the deviations were analyzed via histograms and Gaussian fits.

**Interfacing**

Optical: We glue a fiber array to the PIC using the Nanoscribe IP-S photoresin. We align the fiber array with a 4-axis optimization stage before applying the glue. After retracking the array, we apply the glue and re-align for optimal transmission before curing the resin. To prevent the resin from flowing into the coupling region, we print a barrier onto the bottom of the fiber array using a Nanoscirbe Quantum X and the photoresin IP-S.

Electrical: A carrier PCB was designed to establish an electrical connection to the chip. The chip is glued to the bare copper area in the center and connected to the PCB via bond wires. The input modulation voltages from the RF-DACs enter the board differentially via Samtec ARC6 connectors and are transitioned to single-ended signals using local baluns (Mini-Circuits' TCM2-43X+). All input modulators share a common connection to the bias voltage, which is decoupled with capacitors located directly underneath. The photodiode output currents are routed to SMA connectors at the left edge to be connected to external TIAs. The TIA inputs provide a 50 R path to ground, while the other end of the photodiodes shares a common, locally decoupled bias voltage like the input EAMs. The TIA outputs are connected to a second set of SMA connectors, transitioned to differential signals using the same baluns as for the inputs, and then routed to ARC6 connectors to

be connected to the RF-ADCs. Although not high-speed, the bias and weight voltages use the same type of connector. All traces are impedance-matched to 50 R (single-ended) or 100 R (differential) and groupwise length-matched.

## Data availability

All data is available in the main text and the supplementary materials.

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

## Acknowledgements
The research is funded by: European Union's Horizon 2020 research and innovation programme (grant no. 101017237, PHOENICS project, WP, MK) and the European Union's Innovation Council Pathfinder programme (grant no. 101046878, HYBRAIN project, WP). We acknowledge financial support from Heidelberg University for the publication fee.

## Author contributions
Conceptualization: L.M., J.D., D.S., W.P. and F.P. Methodology: L.M., J.D., S.T., L.M.R., N.B., S.K., J.B., N.G.P. and F.P. Investigation: L.M., J.D., S.T., F.P. and N.B. Visualization: L.M. and J.D. Funding acquisition: D.S., M.K., J.D.J. and W.P. Project administration: W.P., F.P. Supervision: W.P., F.P. Writing—original draft: L.M., F.B., W.P. Writing—review and editing: All authors

## Funding

## Competing interests
The authors declare no competing interests.
