## [Transparent Peer Review file · Nature Communications]

Deep Neural Network Inference on an Integrated, Reconfigurable Photonic Tensor Processor

Corresponding Author: Dr Frank Brückerhoff-Plückelmann

Version 0:

Reviewer comments:

Reviewer #1

(Remarks to the Author)

The manuscript "Deep Neural Network Inference on an Integrated, Reconfigurable Photonic Tensor Processor" presents a photonic accelerator utilizing Electro-absorption Modulators (EAMs) for matrix operations. While the authors demonstrate a system capable of executing a high percentage of the computational workload in the optical domain during the "Deep Neural Network Inference" phase, I have identified critical deficiencies regarding the hardware novelty, device characterization, and the definition of computing precision.

Consequently, I do not believe this work represents the significant technological advance required for publication in Nature Communications. My specific concerns are detailed below:

1. Limited Hardware Innovation:

The proposed architecture lacks sufficient hardware novelty. The fundamental design of the incoherent photonic tensor processor closely resembles architectures that have already been extensively reported in previous literature. Specifically, the hardware concepts presented here do not distinguish themselves significantly from prior works such as [Ref 1: <https://doi.org/10.1038/s41467-024-49768-y>] and [Ref 2: <https://doi.org/10.1038/s41586-024-07590-y>]. Given the established state of the art, the incremental contribution of this specific implementation is limited.

2. Insufficient Characterization and Utilization of EAMs:

There is a notable absence of bandwidth characterization for the Electro-absorption Modulators (EAMs) in the manuscript. While the authors employ EAMs as weight encoding units, they fail to exploit the primary advantage of these devices: high-speed modulation. The manuscript notes that a full weight-array reprogram takes 62 ms, effectively treating the EAMs as slow, quasi-static variable attenuators. In the current landscape of optical computing, performing low-speed weighting on such small-scale optical matrices (10x10) fails to demonstrate the competitive throughput or latency advantages of optical computing compared to electronic baselines.

3. Questionable Definition of Computing Precision:

The authors' reliance on "multiple averages" to define the system's accuracy is problematic. The manuscript states that averaging four repetitions lowers the error to about 11% (precision mode) and further averaging approaches a noise floor. However, statistical averaging is a post-processing technique that trades latency for signal-to-noise ratio; it does not represent the intrinsic, single-shot computational precision of the Photonic Tensor Processor. Presenting averaged results as the hardware's native precision capability masks the true limitations of the analog noise floor and quantization effects.

Conclusion:

Although the demonstration of a high optical workload proportion is a positive integration step, the lack of architectural novelty, the underutilization of the EAMs' high-speed capabilities, and the reliance on averaging to boost precision metrics limit the impact of this work. Therefore, I cannot recommend it for publication in Nature Communications.

Reviewer #2

(Remarks to the Author)

Please receive my review of the paper titled "Deep Neural Network Inference on an Integrated, Reconfigurable Photonic Tensor Processor" authored by L. Meyer et al. The manuscript presents experimental measurement results of a photonic

tensor processor designed to perform artificial neural network inference. The authors demonstrate an integrated system with nine input and three output channels. The system is capable of parallel, intensity-based accumulation of weighted signals using incoherent light.

The manuscript shows a good system demonstration with a high degree of integration. This work presents a fully packaged, rack-mounted system—complete with a Pytorch software stack. The novelty of the work is further enhanced with the use of an optical microcomb as the input light source. The microcomb lends itself to a wavelength-division multiplexed input, where the results of matrix multiplications can be integrated/summed automatically by the output photodetectors. The authors have also included a comprehensive calibration and error analysis of their hardware. Such analysis is useful for other researchers who seek to reproduce the results presented in this manuscript.

Overall, the manuscript is well written, but there are several important technical concerns that should be addressed.

1. How do the authors encode data in the different frequencies/wavelengths of the optical frequency comb?
2. I notice significant input power differences between the different microcomb lines in Figure 2(c). EAMs typically also have strong wavelength dependence where modulation is stronger at lower wavelengths. The authors share some calibration procedure in the supplemental information, but it isn't clear if the procedure successfully addresses this issue. Given the importance of addressing this power variation, the authors should include this in their discussion section.
3. To achieve high accuracy ("precision mode"), the system requires averaging multiple shots/measurements, which inherently reduces the throughput and latency advantages that photonics promises. What is the input/weight/output bit precisions of the system in the single-shot ("low-latency") mode and in the high-accuracy ("precision") mode?
4. The specific implementation described involves nine input and three output channels. This is relatively small compared to the tensor sizes in modern AI models. The authors should address how they would realistically increase the size of photonic tensor processor. The authors should also discuss how convolution operations are performed in the hardware. Representing a convolution operation, even when the kernel is small, can result in large matrix multiplications. When large matrix multiplications are performed by partial smaller multiplications with limited output precision, the resulting precision of the overall multiplication can degrade significantly. The authors should discuss whether they have successfully overcome this issue with measured data.
5. The CIFAR-10 accuracy observed at 72% is very low compared to the performance of the digital hardware, why? Is this due to size or bit precision?
6. The use of the word all-optical: while linear tensor/matrix multiplications are performed optically on chip, the computation of the nonlinear functions are still performed digitally. Therefore, I don't think it's justified to use the term "all-optical" in this context. All-optical deep learning hardware typically performs the nonlinear calculations optically either using atom-light interaction or nonlinear optical components.
7. The paper would benefit heavily from a more direct comparison of energy-per-operation (TOPS/W) against conventional electronics and other optical tensor processors to solidify the "efficiency" claims.

In summary, the manuscript is well written and is technically sound. The authors, however, should address the shortcomings in their manuscript for it to be appropriate for the readership of Nature Communications.

Reviewer #3

(Remarks to the Author)

The paper titled "Deep Neural Network Inference on an Integrated, Reconfigurable Photonic Tensor Processor" demonstrates a notable system-level integration of optical computing. The engineering work is impressive, and the work appears to be within the scope of Nature Communications. However, several key issues require clarification to strengthen the manuscript.

1. The description of the system as an "all-optical photonic tensor processor" may be an overstatement. Even considering the tensor core in isolation, it appears to rely on supporting electronic components for operation.
2. The micro-architecture of the tensor core is not sufficiently detailed. The description largely relies on the term "crossbar" without providing a clear structural diagram. A dedicated schematic illustrating the core's design would significantly improve clarity.
3. Key system-level specifications are missing. These include:
 - a) The clock rates for vector input and matrix-vector multiplication (MVM) output.
 - b) The system's capability for pipelined operation.
 - c) A breakdown of the 63 ms weight loading time, identifying the dominant contributor.
 - d) A power consumption breakdown for the entire system.
 - e) The measured or theoretical peak computational throughput.
4. There is an inconsistency regarding numerical precision. The text mentions 4 bits (Line 185), but Fig. 3c shows approximately 50 distinct weight levels. A clarification on the effective bit-width for both weights and activations is necessary.
5. The x-axis label in Fig. 3a is ambiguous. Given the data points, it likely represents the value of 'n', instead of '2n'. The label should be defined explicitly.
6. The weight error analysis (from Line 396) appears to assume an errorless input vector, which is an unrealistic simplification. The observed photonic output error is likely a function of errors in both the weights and the input vector. This analysis should be detailed to be convincing.
7. For the MNIST and CIFAR-10 classification tasks, it would be valuable to include baseline results obtained from standard electronic hardware (e.g., CPUs/GPUs) for a more direct performance comparison.

Version 1:

Reviewer comments:

Reviewer #2

(Remarks to the Author)

The authors have addressed the technical concerns I previously highlighted in the updated manuscript. I support the publication of this manuscript on Nature Communications. The manuscript shows a good system demonstration with a high degree of integration: from hardware to software.

Minor comments:

I notice the authors only deleted the word "all" from "all-optical" because of my original concern #6. However, it leads to awkward sentences such as in the abstract: "Here we present an optical photonic tensor processor ..." Please revise the grammatical mistakes.

Reviewer #3

(Remarks to the Author)

The authors have provided thoughtful and substantive replies to most of the problems raised and made revisions and supplements in response to the review comments, significantly improving the scientific rigor and clarity of the manuscript. The presented work on an integrated photonic tensor processor demonstrates the feasibility and unique value of optical computing for low-latency, programmable DNN inference. Current limitations, primarily related to engineering maturity (e.g., energy efficiency), are also discussed candidly. This manuscript is acceptable for publication.

Response Letter Deep Neural Network Inference on an Integrated, Reconfigurable Photonic Tensor Processor

Lennart Meyer, Jelle Dijkstra, Simon Tebeck, Liam McRae, Niklas Bahr, Daniel Steinmeyer, Sergey Koptyaev, Johana Bernasconi, Nikolay G. Pavlov, Maxim Karpov, John D. Jost, Wolfram Pernice, Frank Brücknerhoff-Plückelmann

Response to reviewers:

We thank all reviewers for their insightful reviews of our manuscript and their very helpful recommendations on how to improve it. We have followed these recommendations in full, as described in detail below.

Reviewer #1:

The manuscript "Deep Neural Network Inference on an Integrated, Reconfigurable Photonic Tensor Processor" presents a photonic accelerator utilizing Electro-absorption Modulators (EAMs) for matrix operations. While the authors demonstrate a system capable of executing a high percentage of the computational workload in the optical domain during the "Deep Neural Network Inference" phase, I have identified critical deficiencies regarding the hardware novelty, device characterization, and the definition of computing precision. Consequently, I do not believe this work represents the significant technological advance required for publication in Nature Communications. My specific concerns are detailed below:

Our Response:

We thank the reviewer for the careful report and for highlighting key points to strengthen the impact of the manuscript. In the revised version, we added high-speed device measurements, clarified and quantified the computing precision, and expanded the discussion of our training approach that removes the need for per-chip retraining and thus supports deployment at scale.

1. Limited Hardware Innovation:

The proposed architecture lacks sufficient hardware novelty. The fundamental design of the incoherent photonic tensor processor closely resembles architectures that have already been extensively reported in previous literature. Specifically, the hardware concepts presented here do not distinguish themselves significantly from prior works such as [Ref 1: <https://doi.org/10.1038/s41467-024-49768-y>] and [Ref 2: <https://doi.org/10.1038/s41586-024-07590-y>]. Given the established state of the art, the incremental contribution of this specific implementation is limited.

Our Response:

We agree that the underlying incoherent crossbar principle has already been reported in the literature. The primary advance of our work is the transition from a component-level tensor-core demonstration to a deployable inference system, including a fully packaged, rack-

mounted platform with an electronic I/O interface, calibration procedures, and PyTorch integration that allows pretrained networks to be executed on photonic hardware without chip-specific retraining. In addition, we explicitly quantify the hardware's noise statistics and use a Gaussian error abstraction during hardware-aware training, so that training is decoupled from the individual chip instance. Taken together, the end-to-end inference examples provide experimental evidence that photonic hardware-accelerators are functional on a system level and support deployment at scale. Thus, our work differs in scope and demonstrated capability from both cited references, which focus on "perfect linear optics" and fidelity restoration of coherent meshes/crossbars, rather than an end-to-end DNN inference system, software integration, and measured inference workload execution (Ref. 1), and the usage of partial coherent light sources for convolution processing, requiring passing also the train dataset through the photonic processor to capture the error properties of the circuits (Ref. 2). We added an additional paragraph to the introduction and discussion section, clearly positioning our work with respect to the existing literature and highlighting its novelty.

2. Insufficient Characterization and Utilization of EAMs:

There is a notable absence of bandwidth characterization for the Electro-absorption Modulators (EAMs) in the manuscript. While the authors employ EAMs as weight encoding units, they fail to exploit the primary advantage of these devices: high-speed modulation. The manuscript notes that a full weight-array reprogram takes 62 ms, effectively treating the EAMs as slow, quasi-static variable attenuators. In the current landscape of optical computing, performing low-speed weighting on such small-scale optical matrices (10x10) fails to demonstrate the competitive throughput or latency advantages of optical computing compared to electronic baselines.

Our Response:

We thank the reviewer for pointing out the insufficient differentiation between weight-array reprogramming time and input modulation/compute bandwidth. The reported 62 ms corresponds to a full reprogramming of the entire weight array (i.e., updating all weight control voltages, including electronic communication and settling/calibration overhead). During inference, weights are held stationary as long as possible, so the weight reprogramming time is amortized over many input vectors and is not incurred per MVM. This has the benefit of reducing digital data shuffling in the system and thus improves efficiency. In contrast, the input activation modulation in our system is operated at 4 GS/s per channel, while the ADCs sample at 2 GS/s per channel, enabling high-rate streaming of input vectors. The current inference-rate ceiling is therefore set by system-level I/O. We added characterization measurements of the EAMs up to 8 GBaud to show the further potential of the photonic devices. Even though the EAMs support operation at higher bandwidth, we set the operation speed of the system to 4 GS/s and 2 GS/s for DACs and ADCs respectively. While higher sampling rates, e.g. 10-20 GS/s, do not have a large impact on the energy per conversion step at the DAC, ADC efficiency drops significantly [1–3]. We clarified the programming times in the manuscript and added a paragraph about the sampling rates in the discussion section.

[1] B. Murmann ADC Performance Survey 1997-2025

[2] Luo L, Chen S, Zhou M, Ye T (2017) A 0.014mm² 10-bit 2GS/s time-interleaved SAR ADC with low-complexity background timing skew calibration. In: 2017 Symposium on VLSI Circuits. IEEE, Kyoto, Japan, pp C278–C279

[3] Li R, Fu J, Ma S, Wu D, Guo X, Wu J (2023) 10GS/s 10bit Time-interleaved SAR ADC in 28nm CMOS. In: 2023 8th International Conference on Integrated Circuits and Microsystems (ICICM). IEEE, Nanjing, China, pp 16–20

3. Questionable Definition of Computing Precision:

The authors' reliance on "multiple averages" to define the system's accuracy is problematic. The manuscript states that averaging four repetitions lowers the error to about 11% (precision mode) and further averaging approaches a noise floor. However, statistical averaging is a post-processing technique that trades latency for signal-to-noise ratio; it does not represent the intrinsic, single-shot computational precision of the Photonic Tensor Processor. Presenting averaged results as the hardware's native precision capability masks the true limitations of the analog noise floor and quantization effects.

Our Response:

We agree that the single shot error of $(19.4 \pm 0.5) \%$ is the intrinsic error of the photonic system and not the four times averaged readout with an error of $(10.9 \pm 0.3) \%$. The noise in the photonic system is dominantly stochastic and not systematic as shown in Fig 3. a, as MVM error decreases to the (systematic) error floor with additional averaging, enabling a perspective for improvement by optimizing the ratio between photonic signal strength and electronic noise from the transimpedance amplifier. We clarified the paragraph in the manuscript, explicitly stating the MVM errors and the throughput/latency penalty induced by averaging.

Conclusion:

Although the demonstration of a high optical workload proportion is a positive integration step, the lack of architectural novelty, the underutilization of the EAMs' high-speed capabilities, and the reliance on averaging to boost precision metrics limit the impact of this work. Therefore, I cannot recommend it for publication in Nature Communications.

Our Response:

We hope these additional measurements and clarifications address the reviewer's concerns.

Reviewer #2:

Please receive my review of the paper titled "Deep Neural Network Inference on an Integrated, Reconfigurable Photonic Tensor Processor" authored by L. Meyer et al. The manuscript presents experimental measurement results of a photonic tensor processor designed to perform artificial neural network inference. The authors demonstrate an integrated system

with nine input and three output channels. The system is capable of parallel, intensity-based accumulation of weighted signals using incoherent light.

The manuscript shows a good system demonstration with a high degree of integration. This work presents a fully packaged, rack-mounted system—complete with a Pytorch software stack. The novelty of the work is further enhanced with the use of an optical microcomb as the input light source. The microcomb lends itself to a wavelength-division multiplexed input, where the results of matrix multiplications can be integrated/summed automatically by the output photodetectors. The authors have also included a comprehensive calibration and error analysis of their hardware. Such analysis is useful for other researchers who seek to reproduce the results presented in this manuscript.

Overall, the manuscript is well written, but there are several important technical concerns that should be addressed.

Our Response:

We thank the reviewer for the positive comments and the detailed technical feedback, which helped improve the manuscript. We revised the manuscript as described in the following:

1. How do the authors encode data in the different frequencies/wavelengths of the optical frequency comb?

Our Response:

We demultiplex the different carriers of the frequency comb and route the individual carriers to the input grating couplers of the SOI PIC. The input EAMs in each arm modulate the data onto the carriers. The input encoded and weighted signals are then accumulated along the different bus waveguides for each row. We clarified the corresponding paragraph in the manuscript.

2. I notice significant input power differences between the different microcomb lines in Figure 2(c). EAMs typically also have strong wavelength dependence where modulation is stronger at lower wavelengths. The authors share some calibration procedure in the supplemental information, but it isn't clear if the procedure successfully addresses this issue. Given the importance of addressing this power variation, the authors should include this in their discussion section.

Our Response:

We thank the reviewer for pointing this out. The microcomb line-to-line power variation and the wavelength-dependent electro-absorption response of the EAMs are important nonidealities that must be compensated to achieve accurate MVM operation. In our system, these effects appear as channel-dependent, nonlinear weight transfer functions, which we explicitly measure and correct using a per-channel weight-map calibration. Specifically, for each channel we sweep the relative range $r \in [-1,1]$ while holding other channels at zero and record the corresponding optical response, yielding a measured calibration curve $f_c: r \mapsto y$ (SI Fig. 9). These curves capture both the different optical powers of the individual comb lines and

the wavelength dependence of the EAM modulation depth. During weight programming, desired weights are mapped into a common feasible target range, and the required drives are obtained by numerically inverting the measured per-channel calibration curves via interpolation ($\rho_c = f_c^{-1}(t_c)$). Then, the measured crosstalk matrix is applied as an additional correction step. We clarified the explanation in the Supplementary Information and expanded the error discussion in the main text.

3. To achieve high accuracy ("precision mode"), the system requires averaging multiple shots/measurements, which inherently reduces the throughput and latency advantages that photonics promises. What is the input/weight/output bit precisions of the system in the single-shot ("low-latency") mode and in the high-accuracy ("precision") mode?

Our Response:

We agree that averaging reduces the throughput and latency of the photonic system. We clearly state this penalty in the main text now. We added a new simulation in the supplementary information, computing the MVM error for a digital system when quantizing the inputs to 8 bit and different weight quantization. The MVM error of $(19.4 \pm 0.5) \%$ is close to a digital system with 8 bit input quantization and 3 bit weight quantization, while the MVM error of $(10.9 \pm 0.3) \%$ is close to 4 bit weight quantization.

4. The specific implementation described involves nine input and three output channels. This is relatively small compared to the tensor sizes in modern AI models. The authors should address how they would realistically increase the size of photonic tensor processor. The authors should also discuss how convolution operations are performed in the hardware. Representing a convolution operation, even when the kernel is small, can result in large matrix multiplications. When large matrix multiplications are performed by partial smaller multiplications with limited output precision, the resulting precision of the overall multiplication can degrade significantly. The authors should discuss whether they have successfully overcome this issue with measured data.

Our response:

Matrix tiling is necessary for most modern AI models with the present 9-input/3-output implementation. We added an outlook for scaling the size and throughput of the system in the main text. Also, hardware software codesign is an appealing approach to tackle this problem. For example, by using heavily grouped convolutions, as the depthwise separable ones in MobileNet, effectively reducing the matrix size. Composing a large MVM from K partial MVMs can indeed degrade precision. For not correlated stochastic noise, the absolute error of a composed output obtained by summing K partial MVMs increases approximately as \sqrt{K} . At the same time, the magnitude of the accumulated signal also increases as \sqrt{K} (for uncorrelated inputs), implying that the relative stochastic error remains approximately constant with tiling. However, for correlated / systematic errors the total error of the tiled MVM increases like K , eventually leading to an increasing relative error. As the noise in our system is mostly stochastic, also see Fig 3 a, the tiling is less problematic. However, it can get a problem for tiling very large matrices as for the CIFAR-10 network. We added a paragraph regarding the error in the results and discussion of the main text.

5. The CIFAR-10 accuracy observed at 72% is very low compared to the performance of the digital hardware, why? Is this due to size or bit precision?

Our response:

The lower accuracy on CIFAR-10 is likely driven by the much larger input-vector dimensions in the network, most notably in the final fully connected layer, where tiling becomes more severe and the accumulation of non-stochastic errors can limit performance. The largest convolution is a 3×3 layer with 64 channels, corresponding to an input vector of 576, whereas the fully connected layer operates on the flattened 8×8×128 tensor, i.e., an input vector of 8192. To isolate this effect, we used the experimentally measured activations from the photonic chip up to (and including) the layer preceding the classifier and executed only the final fully connected layer digitally. This increased the accuracy from 72% to 78%, supporting the conclusion that tiling and the associated non-stochastic error accumulation are a limiting factor at larger matrix sizes. We thank the reviewer for prompting this discussion.

6. The use of the word all-optical: while linear tensor/matrix multiplications are performed optically on chip, the computation of the nonlinear functions are still performed digitally. Therefore, I don't think it's justified to use the term "all-optical" in this context. All-optical deep learning hardware typically performs the nonlinear calculations optically either using atom-light interaction or nonlinear optical components.

Our response:

We agree that the system is not "all-optical" in the strict sense. While the linear operations are performed in the optical domain, nonlinear activations, control, and calibration are implemented electronically. We originally used "all-optical" to emphasize that the linear accumulation is carried out optically (in contrast to approaches that use optical encoding but perform weighting and/or accumulation electronically, e.g. <https://doi.org/10.1038/s41586-025-08786-6> and <https://doi.org/10.1038/s41586-025-08854-x>). To avoid ambiguity, we have changed the phrasing in main text.

7. The paper would benefit heavily from a more direct comparison of energy-per-operation (TOPS/W) against conventional electronics and other optical tensor processors to solidify the "efficiency" claims.

Our response:

We thank the reviewer for the suggestion and added a complete power consumption breakdown in the Supplementary Information and a paragraph in the discussion of the main text comparing the PTP to other analog processors and state of the art GPUs.

In summary, the manuscript is well written and is technically sound. The authors, however, should address the shortcomings in their manuscript for it to be appropriate for the readership of Nature Communications.

Our Response:

We thank the reviewer for the positive evaluation and for raising the technical points above. We believe the revisions address these concerns and improve the clarity and completeness of the manuscript

Reviewer #3:

The paper titled "Deep Neural Network Inference on an Integrated, Reconfigurable Photonic Tensor Processor" demonstrates a notable system-level integration of optical computing. The engineering work is impressive, and the work appears to be within the scope of Nature Communications. However, several key issues require clarification to strengthen the manuscript.

Our Response:

We thank the reviewer for acknowledging our work and for the in-depth, which helped us further strengthen the manuscript. We revised the manuscript as described in the following:

1. The description of the system as an "all-optical photonic tensor processor" may be an overstatement. Even considering the tensor core in isolation, it appears to rely on supporting electronic components for operation.

Our Response:

We thank the reviewer for raising this point. The major difference to other "optical" implementation like <https://doi.org/10.1038/s41586-025-08786-6> and <https://doi.org/10.1038/s41586-025-08854-x> is that all parts, broadcasting of the inputs, weighting and accumulation are implemented in the optical domain. We have changed the phrasing in the main text to clarify this point.

2. The micro-architecture of the tensor core is not sufficiently detailed. The description largely relies on the term "crossbar" without providing a clear structural diagram. A dedicated schematic illustrating the core's design would significantly improve clarity.

Our Response:

We agree with the reviewer and added a diagram in the supplementary information to explain the crossbar architecture.

3. Key system-level specifications are missing. These include:

Our Response:

We thank the reviewer for pointing this out. We added the following specifications to the manuscript.

- a) The clock rates for vector input and matrix-vector multiplication (MVM) output.

The ADCs and DACs run at 4 GSa/s and 2 GSa/s respectively. With the four-samples per input encoding and two-sample read out scheme the operation rate of the system is 1 GHz. We clarified this in the method part of the main text

b) The system's capability for pipelined operation.

One main advantage of photonic data processing is that the computation itself is implemented by the propagation of optical fields within the integrated circuit. In contrast to systolic arrays, the full matrix vector multiplication is implemented in a single symbol period / "clock cycle" and no pipelining is required, drastically reducing latency. While the second symbol can in principle propagate through the photonic circuit before the previous is detected at the photodetector, the propagation time through the circuit is diminishing small in comparison through the symbol length for the given photonic circuit on a sub 1 mm scale. We added this information in the discussion part of the main text.

c) A breakdown of the 63 ms weight loading time, identifying the dominant contributor.

Our system is operated in a weight-stationary inference regime, where weights are kept fixed for long sequences of input vectors/patches to minimize digital data transfer. The measured 62 ms corresponds to a full-array update and is dominated by control-path overhead (FPGA evaluation-board communication), rather than by the intrinsic response time of the EAMs or the update capability of the DAC controlling the weights. The LTC2668 DACs would support microsecond-scale update capability.

d) A power consumption breakdown for the entire system.

We added a complete Power consumption breakdown in the Supplementary Information. The system currently consumes 17.78 W and has a projected power consumption of 2.5 W.

e) The measured or theoretical peak computational throughput.

The computational throughput of the photonic tensor core under continuous streaming operation with an effective symbol rate of 1 GHz and a 9×3 MVM per symbol is 27 GMAC/s. Using the standard convention 1 MAC = 2 operations (multiply + accumulate), this corresponds to a peak throughput of 0.054 TOPS in the present system configuration (single-shot, weights stationary, no averaging). The intrinsic opto-electronic devices (EAMs and photodetectors) support bandwidths in the tens of GHz, if the system-level electronics and packaging were upgraded to operate at 50 GHz symbol rates, the corresponding peak core throughput would scale proportionally to ~ 2.7 TOPS. We added this information, in addition to a general scaling outlook, to the discussion part of the main text.

4. There is an inconsistency regarding numerical precision. The text mentions 4 bits (Line 185), but Fig. 3c shows approximately 50 distinct weight levels. A clarification on the effective bit-width for both weights and activations is necessary.

Our Response:

We thank the reviewer for raising this point. While the number of distinct weight levels is an important factor, it is not the only contribution to overall precision. We therefore compare our measured MVM error to the error of an ideal digital MVM with 8-bit input quantization and varying weight quantization: $(19.4 \pm 0.5)\%$ is comparable to 3-bit weights and $(10.9 \pm 0.3)\%$ to 4-bit weights. We added a new section in the Supplementary Information simulating the equivalent digital weight quantization. The DACs and ADCs feature 14-bit and we added new measurements on the full DAC-EAM-PD-ADC transmission path to the supplementary information. The systematic transmission error of 0.3 % is small in comparison to the systematic MVM error of the full system, indicating that it's not constraint by the input/output interface.

5. The x-axis label in Fig. 3a is ambiguous. Given the data points, it likely represents the value of 'n', instead of '2n'. The label should be defined explicitly.

Our Response:

We agree with the reviewer and changed the x-axis label of Fig. 3a.

6. The weight error analysis (from Line 396) appears to assume an errorless input vector, which is an unrealistic simplification. The observed photonic output error is likely a function of errors in both the weights and the input vector. This analysis should be detailed to be convincing.

Our Response:

We agree that the observed output error can arise from both weight and input errors, and that finite ENOB and residual non-linearities (EAM transfer function, PD/TIA) can contribute. In the revised manuscript/Supplementary Information, we clarify this point and include the additional DAC-EAM-PD-ADC transmission measurement described above. The systematic contribution from the I/O chain is small (systematic transmission error of 0.3 %) compared to the MVM error.

7. For the MNIST and CIFAR-10 classification tasks, it would be valuable to include baseline results obtained from standard electronic hardware (e.g., CPUs/GPUs) for a more direct performance comparison.

Our Response:

The performance of the network strongly depends on the chosen architecture. For the networks used in the manuscript and shown in Table 1, the accuracy for MNIST is 98.9 % and for CIFAR 10 89.5 %. Using advanced vision transformer models, accuracies above 99 % are possible for both datasets, <https://doi.org/10.48550/arXiv.2205.12755> .

We thank the editor and reviewers for their time and constructive feedback.

Response Letter Deep Neural Network Inference on an Integrated, Reconfigurable Photonic Tensor Processor

Lennart Meyer, Jelle Dijkstra, Simon Tebeck, Liam McRae, Niklas Bahr, Daniel Steinmeyer, Sergey Koptyaev, Johana Bernasconi, Nikolay G. Pavlov, Maxim Karpov, John D. Jost, Wolfram Pernice, Frank Brückerohoff-Plückelmann

Reviewer #2 (Remarks to the Author):

The authors have addressed the technical concerns I previously highlighted in the updated manuscript. I support the publication of this manuscript on Nature Communications. The manuscript shows a good system demonstration with a high degree of integration: from hardware to software.

Minor comments:

I notice the authors only deleted the word "all" from "all-optical" because of my original concern #6. However, it leads to awkward sentences such as in the abstract: "Here we present an optical photonic tensor processor ..." Please revise the grammatical mistakes.

Response to reviewers:

We appreciate the careful revision of our manuscript, and we corrected the grammatical mistake. We thank the reviewer for his detailed feedback during the review process and for supporting the publication of our work.

Reviewer #3 (Remarks to the Author):

The authors have provided thoughtful and substantive replies to most of the problems raised and made revisions and supplements in response to the review comments, significantly improving the scientific rigor and clarity of the manuscript. The presented work on an integrated photonic tensor processor demonstrates the feasibility and unique value of optical computing for low-latency, programmable DNN inference. Current limitations, primarily related to engineering maturity (e.g., energy efficiency), are also discussed candidly. This manuscript is acceptable for publication.

Response to reviewers:

We once again thank the reviewer for the thoughtful feedback, which has significantly improved the quality of our work. We also appreciate that the reviewer is satisfied with our previous responses.